# A Water Balancing Act: Water Balances Highlight the Benefits of Community-Based Adaptive Management in Northern New Mexico, USA

Lily M. Conrad [1], Alexander G. Fernald [1,*], Steven J. Guldan [2] and Carlos G. Ochoa [3]

1    Water Resources Research Institute, New Mexico State University, Las Cruces, NM 88003, USA; conradl@nmsu.edu
2    Sustainable Agriculture Science Center, New Mexico State University, Alcalde, NM 87511, USA; sguldan@nmsu.edu
3    Ecohydrology Laboratory, College of Agricultural Sciences, Oregon State University, Corvallis, OR 97331, USA; carlos.ochoa@oregonstate.edu
*    Correspondence: afernald@nmsu.edu; Tel.: +1-575-646-4337

**Abstract:** Quantifying groundwater recharge from irrigation in water-scarce regions is critical for sustainable water management in an era of decreasing surface water deliveries and increasing reliance on groundwater pumping. Through a water balance approach, our study estimated deep percolation (*DP*) and characterized surface water and groundwater interactions of two flood-irrigated fields in northern New Mexico to evaluate the regional importance of irrigation-related recharge in the context of climate change. *DP* was estimated for each irrigation event from precipitation, irrigation input, runoff, change in soil water storage, and evapotranspiration data for both fields. Both fields exhibited positive, statistically significant relationships between *DP* and total water applied (TWA), where one field exhibited positive, statistically significant relationships between *DP* and groundwater level fluctuation (GWLF) and between GWLF and total water applied. In 2021, total *DP* on Field 1 was 739 mm, where 68% of irrigation water applied contributed to *DP*. Field 2′s total *DP* was 1249 mm, where 81% of irrigation water applied contributed to *DP*. Results from this study combined with long-term research indicate that the groundwater recharge and flexible management associated with traditional, community-based irrigation systems are the exact benefits needed for appropriate climate change adaptation.

**Keywords:** flood irrigation; water management; deep percolation; surface water; groundwater; water balance

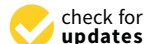



## 1. Introduction

Over 50% of the world's freshwater resources for human use and consumption rely on river discharge that can be greatly impacted by long-term changes in precipitation and temperature such as those caused by climate change, particularly in snow-dominated regions [1]. Much of the western United States depends on precipitation falling in the winter in mountainous regions as snow and subsequently released slowly as snowmelt throughout the following spring and summer seasons. However, long-term changes in temperature and precipitation are already affecting these crucial water resource systems by decreasing the maximum snowpack accumulation, shifting the timing of runoff to arrive earlier, and impacting the volume of river discharge [2,3] with changes amplified by a lack of reservoir storage [4]. More specifically, snow-dominated basins in the mid-high latitudes are the most vulnerable to the impacts of warming climates where maximum runoff is expected to arrive one month earlier by 2050 in the western United States [1].

For example, the Rio Grande and its tributaries are increasingly becoming water stressed due to the warming climate and the increasing demand from users in Colorado, New Mexico, Texas, and Mexico [5]. Rio Grande streamflow is vulnerable as it largely

depends on snowpack conditions which are projected to decrease and melt earlier in the future [6–9]. This surface water resource must serve industrial, tourist, residential, agricultural, ecologic, and economic needs in the USA (e.g., Colorado, New Mexico, Texas) and Mexico. Under current climate conditions, New Mexico does not have water to spare between all users [10].

Oftentimes, agricultural sectors are the largest users of water and face greater pressure to develop new water management strategies to help non-agricultural sectors cope with future water scarcity caused by warming temperatures and climate uncertainty [11,12]. In New Mexico in 2015, irrigated agriculture accounted for 76% of total water use, 53% from surface water and 47% from groundwater. Flood irrigation is used on 45% of all irrigated fields in New Mexico [13].

Common water delivery systems for flood irrigation in New Mexico are *acequia* networks which face many socio-environmental challenges. First introduced to northern New Mexico in the 16th century, acequias are gravity-driven water delivery networks and also serve as the basis of community-managed water governance systems [14,15]. While acequias have many beneficial hydrologic (e.g., aquifer recharge) and social attributes (e.g., water sharing) that foster resilience [16], these ancient water systems still face the challenge of long-term, regional drought and difficult water policy [17,18]. Questions are continually raised at acequia irrigator meetings and posed to researchers regarding what the "right" management strategies are: Should we line the canals? Should we switch to drip? Should we pump groundwater? Irrigators find themselves stuck between cultural norms of propagating generational knowledge of traditional irrigation methods and pressures from decreasing water availability and outside agencies to modernize water delivery systems and maximize irrigation efficiency.

Agricultural irrigation practices involving surface water can cause percolation and groundwater recharge that significantly impact groundwater resources on regional scales [12,19–22]. A study by Bouimouass et al. (2020) focused on the acequia counterpart in Morocco—*seguias*—and concluded that flood irrigation of diverted surface water resulted in the dominant recharge process in mountain front landscapes [23]. Other studies from large agricultural drainages in China found that approximately 70% of applied flood irrigation water in maize fields recharged the groundwater during the growing season [24], and seepage from both irrigation canals and deep percolation (*DP*) from irrigation contributed to more than 90% of total annual shallow groundwater recharge [12]. Additionally, in a large traditional agricultural basin in Italy, irrigation water delivered through a system of canals provided 55 to 88% of groundwater recharge [25]. *DP* is the amount of water that travels below the effective root zone (ERZ) that can potentially reach the shallow aquifer [26]. One of our previous studies conducted in northern New Mexico showed that peak groundwater level response fluctuated up to 380 mm 8 to 16 h after the onset of flood irrigation [27], where another estimated 16% of unlined irrigation canal flows seeped into the subsurface, causing the water table to rise 1 to 1.2 m [28]. Additionally, annual shallow aquifer recharge ranged from 1044 to 1350 mm on a valley scale [22]. In these cases, *DP* from flood irrigation was a significant source of recharge to shallow groundwater. *DP* below the vegetative root zone can provide very important hydrologic and ecosystem benefits in irrigated valleys of semiarid and arid regions.

Conversely, groundwater may display evidence of interactions with surface water. As irrigation water infiltrates into the shallow aquifer, this *DP* can contribute groundwater return flows to the river. In northern New Mexico, this interaction is of particular interest considering *DP* can serve as temporary subsurface storage which provides delayed return flow during low-flow periods [22,29,30]. This serves as an important possible buffer for changing peak runoff timing associated with climate variability [19].

Considering interacting surface water and groundwater as one resource is essential for optimal protection of watersheds, sustaining water resources, and furthering integrated groundwater management [20,31]. This is critical within irrigation districts that are increasingly relying on pumping groundwater for agricultural and municipal uses, which can lead

to the disconnection of surface water and groundwater [32]. More recently, groundwater recharge via flooding fields is becoming a more common conservation practice [33,34].

It is necessary to properly quantify aquifer recharge and foster an accurate understanding of *DP* and surface water and groundwater interactions in water-limited regions [35]. The water balance method is a technique commonly used to quantify groundwater recharge and characterize surface water and groundwater interactions [19,22,26,36,37]. Components of the water balance are precipitation, irrigation water applied, runoff, change in soil water storage, and evapotranspiration, where *DP* is unknown and calculated by the difference of these inputs and outputs [26].

Our first objective was to characterize and compare surface water and groundwater interactions and shallow aquifer response to irrigation events in flood-irrigated forage grass fields located within the same irrigated valley in northern New Mexico by estimating *DP* below the root zone with a water balance approach. Our second objective was to justify community-based adaptive management in the context of climate change by relating field-scale findings to regional climate change literature. The innovative approach of identifying tightly coupled objectives reflected the unique, tightly coupled natural and human irrigation system our study focused on. While cultivating a better understanding of available surface water resources is extremely important, irrigators and policy makers must also understand the effects of irrigation techniques on groundwater and surface water availability for downstream users [31]. Previous studies have quantified and compared *DP* across several crop fields, soil types, and valleys in northern New Mexico, USA [22,26]; however, more field observations of *DP* are needed to expand these studies from field-scale to valley or regional scales. We hypothesized that: (1) *DP* and total water applied and *DP* and groundwater response would be positively related on both fields; and (2) *DP* and total water applied would be significantly different across both study fields.

## 2. Materials and Methods

### 2.1. Study Area

This study was conducted on two acequia-irrigated fields in the Rio Hondo agricultural valley in northern New Mexico, USA. The Rio Hondo watershed drains an area of 185 km$^2$ [38] and is located 2200 m above sea level [18]. Snowmelt from the Sangre de Cristo Mountains serves as the Rio Hondo's main source of water and drains to the Rio Grande. Located in a semiarid steppe climate, 50% of the precipitation in this region falls during the monsoon season from June to September [39] with an annual average of 300 mm·year$^{-1}$ [40]. The primary settlements in the Rio Hondo Valley are Valdez and Arroyo Hondo. The agricultural activity in the Rio Hondo watershed is small-scale in nature. Eight canals divert water from the Rio Hondo and deliver irrigation water to approximately 1161 ha through a system of branching acequias [38]. Typical crops include grasses (Mostly *Phleum pretense, Poa pratensis*), alfalfa (*Medicago sativa)*, orchards (e.g., plums, apples, apricot), and vegetables (e.g., squash, beets, greens, onions, radishes, etc.) [41].

Located in the community of Valdez within the Rio Hondo watershed, the first study field (F1) was approximately 27 km north of Taos, New Mexico, USA and covered 2.51 ha (Figure 1a). The main crops grown on the field were grasses, alfalfa, and clovers. The second study field (F2), located in the community of Arroyo Hondo within the Rio Hondo watershed, was approximately 20 km north of Taos, New Mexico (Figure 1b) and covered 1.62 ha. The main crops growing were grasses, alfalfa, and clovers.

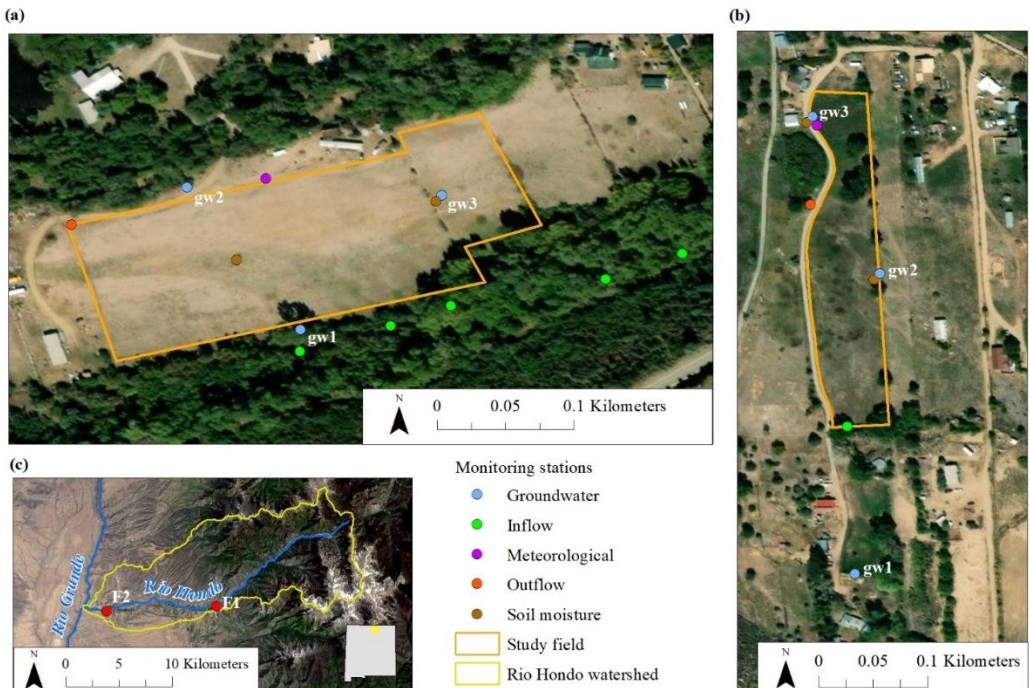

**Figure 1.** Water balance field study sites: (**a**) F1 (located at 36°32′05.3″ N, 105°34′04.5″ W); (**b**) F2 (located at 36°31′47.8″ N, 105°41′00.7″ W) and the corresponding monitoring stations. Both fields are located in the Rio Hondo watershed in Taos County, northern New Mexico ((**c**) inset). Monitoring station locations were selected to most accurately represent average field conditions of the irrigated area while also considering landowner needs for equipment maneuverability while cutting hay.

Soil Physical Properties

Of F1′s total 2.51 ha, 2.35 ha were Manzano clay loam, and 0.16 ha were Loveland clay loam soil types. For the Manzano clay loam soil, slope values typically range from 3 to 5%, the soil is well-drained with medium runoff, average depth to the water table is more than 2 m, and a typical soil profile is clay loam for the top 1.5 m. For the Loveland clay loam soil, slope values typically range from 0 to 3%, the soil drains poorly and has a high runoff class, average depth to the water table is 0.15 to 0.46 m, and a typical soil profile is clay loam for the top 0 to 0.23 m, sandy clay loam for the middle 0.23 to 0.53 m, and very gravelly sand for the bottom 0.53 to 1.52 m [42].

Of F2′s total 1.62 ha, 1.29 ha were Fernando silt loam, 0.24 ha were Fernando clay loam, and 0.08 ha were from the Sedillo–Silva association. The Fernando silt loam slope values typically range from 0 to 7% and are well drained with medium runoff. The average depth to the water table is greater than 2 m, and a typical soil profile is silt loam for the top 0 to 0.20 m, silty clay loam for the middle 0.20 to 0.91 m, and silt loam for the bottom 0.91 to 1.52 m. The Fernando clay loam generally has slope values from 3 to 5%, is well drained with medium runoff, depth to the water table is more than 2 m, and a typical soil profile is clay loam for the top 0 to 0.18 m, silty clay loam for the middle 0.18 to 0.64 m, and silty loam for the bottom 0.64 to 1.52 m. The Sedillo–Silva association soil typically has a slope of 10 to 25%, is well drained with high runoff, depth to the water table is greater than 2 m, and a typical soil profile is very gravelly loam for the top 0 to 0.08 m, gravelly clay loam for the middle 0.08 to 0.28 m, and very cobbly sandy loam for the bottom 0.28 to 1.52 m [43].

Soil bulk density varied between the two fields, whereas soil texture remained relatively consistent (Table 1). For F1, bulk density ranged from $1.46 \times 10^9$ Mg·m$^{-3}$ in the topsoil to $1.23 \times 10^9$ Mg·m$^{-3}$ toward the bottom of the soil profile. Within the field F2 soil profile, bulk density ranged from $1.19 \times 10^9$ Mg·m$^{-3}$ to $1.27 \times 10^9$ Mg·m$^{-3}$ from top to bottom. Soil texture was sandy clay loam for all soil depths except the top layer of the F1 soil profile which was sandy loam. Soil texture components exhibited the same trends

through the soil profile for sand and silt but differed for clay. Sand content decreased, and silt content increased toward the bottom of the soil profile for both fields, while clay content increased in F1 and decreased in F2 (Table 1).

**Table 1.** Soil physical properties for the two field sites from manual soil sample collection (see Section 2.2.3). Laboratory analysis determined soil bulk density, soil particle distribution, and soil texture for each sensor depth in the soil profile. Values for each soil depth represent the averaged value between the two soil-monitoring stations on each field.

| Field | Soil Depth (m) | Bulk Density $(Mg \cdot m^{-3})$ | Sand (%) | Silt (%) | Clay (%) | Soil Texture |
|---|---|---|---|---|---|---|
| F1 | 0.2 | $1.46 \times 10^9$ | 72.6 | 17.6 | 9.90 | Sandy loam |
| | 0.5 | $1.35 \times 10^9$ | 55.6 | 27.5 | 16.9 | Sandy clay loam |
| | 0.8 | $1.23 \times 10^9$ | 51.7 | 31.5 | 16.9 | Sandy clay loam |
| F2 | 0.2 | $1.19 \times 10^9$ | 60.6 | 27.4 | 12.0 | Sandy clay loam |
| | 0.5 | $1.26 \times 10^9$ | 59.6 | 32.5 | 7.90 | Sandy clay loam |
| | 0.8 | $1.27 \times 10^9$ | 57.5 | 32.4 | 10.0 | Sandy clay loam |

*2.2. Field Data Collection*

We monitored various parameters at both study sites to calculate *DP* using a water balance approach for irrigation events over the 2020 and 2021 irrigation seasons. The water balance method was an appropriate approach for our study given our goals of estimating recharge for individual irrigation events within an irrigation season and subsequently relating our findings to community adaptive management and climate change. *DP* is the water that infiltrates into the subsurface, past the ERZ. ERZ varies depending on crop root development, effective soil depth, soil fertility or fertility management, and soil physical properties [44]. We recorded ERZ measurements of root systems at each site during soil volumetric water content (*θ*) sensor installation where F1 ERZ was 0.51 m and F2 ERZ was 0.53 m. Data collected throughout the 2020 and 2021 irrigation seasons returned a groundwater recharge estimate for each irrigation event through a field-scale water balance approach:

$$DP = PPT + IRR - RO - \Delta S - AET \tag{1}$$

where *PPT* is the amount of rainfall during the time interval (mm), *IRR* is irrigation water applied during the time interval (mm), *RO* is the amount of irrigation runoff during the time interval (mm), *ΔS* is the change of storage or change in *θ* during the time interval (mm), and *AET* is the actual evapotranspiration during the time interval (mm). The time interval for each irrigation event begins with the onset of irrigation and extends to 24 h after the end of the irrigation water delivery to achieve an assumed state of field capacity.

2.2.1. Precipitation

Precipitation falling on the study sites during the irrigation season was mainly rainfall measured by weather stations on each field. Both weather stations were equipped with a tipping bucket rain gauge (ClimaVUE50, Campbell Scientific, Inc.; Logan, UT, USA) programmed to record incremental precipitation every five minutes.

2.2.2. Irrigation Inflow and Outflow

Property owners and field managers for both fields used acequia-delivered surface water to flood irrigate throughout the growing season and decided to irrigate based on water allocations, environmental conditions, and crop needs. Surface water is diverted from the acequia onto fields through a series of wooden and metal headgates depending on the size and orientation of the field with respect to the acequia. F1 had five irrigation inflow monitoring stations and one irrigation outflow station. F2 had one irrigation inflow monitoring station and one irrigation outflow station. Rectangular Samani–Magallanez flumes [45] installed at inflow and outflow locations, each equipped with a CS451 pres-

sure transducer and a CR300 datalogger (Campbell Scientific, Inc.; Logan, UT, USA) and programmed to record water level at five-minute increments measured *IRR* and *RO* on each field.

### 2.2.3. Soil Water Content and Physical Properties

Derived from soil volumetric water content data, the change in storage was determined as:

$$\Delta S = \sum_{i=1}^{n} (\theta_2 - \theta_1)_i \Delta d_i \tag{2}$$

where $n$ is the number of layers represented by a soil sensor in the ERZ profile, $\theta_1$ is the soil volumetric water content at the onset of irrigation ($m^3 \cdot m^{-3}$), $\theta_2$ is the soil volumetric water content 24 h after irrigation ends or the average soil volumetric water content at field capacity ($m^3 \cdot m^{-3}$), and $\Delta d_i$ is the soil layer thickness (mm). Equation (2) converted $\theta$ at each sensor location to the amount of water (mm) held in the ERZ.

Each field had two monitoring stations measuring $\theta$. At each station, a CR300 datalogger and three horizontally placed CS655 (Campbell Scientific, Inc.; Logan, UT, USA) soil sensors were arranged vertically in the ERZ at depths of 0.2 m, 0.5 m, and 0.8 m and recorded changes in $\theta$ every minute and averaged data at 30-min increments.

Soil samples collected while installing the sensor network underwent laboratory analysis to determine soil texture and bulk density. Three soil cores were collected at each sensor depth on the opposite wall of the pit where sensors were installed with a split soil core sampler and analyzed with the Blake and Hartge bulk density method [46] and the Gee and Bauder hydrometer method to determine soil texture [47].

### 2.2.4. Evapotranspiration

We used the following equation to calculate the amount of actual evapotranspiration (*AET*):

$$AET = K_c ET_0 \tag{3}$$

where $ET_0$ is the total evapotranspiration (mm) calculated with the Penman–Monteith equation programmed into a CR1000 datalogger (Campbell Scientific, Inc.; Logan, UT, USA). The Penman–Monteith equation outperforms others by including more factors that influence crop water loss (e.g., absorbed radiant energy, wind, atmospheric vapor deficit) and is therefore expected to provide more accurate estimates [48]. Post-processing the $ET_0$ values with crop coefficient ($K_c$) values calculates *AET*. We used crop coefficient curves for grass at different stages of the growing season presented in a previous study that took place near our study area [49] (p. 151). $ET_0$ values were recorded, and *AET* values were calculated for hourly data.

### 2.2.5. Groundwater Level

Three monitoring wells equipped with water level loggers (HOBO Logger U20-001-01, Onset; Bourne, MA, USA) recorded water table fluctuations on each field. All monitoring wells on F1 were steel drive-point wells 2 to 3 m deep. Two of these wells were installed by previous researchers [26]. We installed two steel drive-point wells 2 to 3 m deep on F2 and used the landowner's residential drinking well that was 13 m deep for the third monitoring well. This residential well has been used for long-term groundwater monitoring, where the data clearly show groundwater level response to the irrigation season.

The groundwater level data helped characterize shallow aquifer response to *DP* from irrigation inputs. Calculated for each irrigation event, groundwater level fluctuation (GWLF) (mm) was the difference between groundwater level prior to the irrigation onset (averaged over the 6 h prior to the irrigation onset) and maximum water level rise until the following irrigation event. Negative GWLF values indicate declining groundwater levels.

### 2.3. Statistical Analyses

Specific parameters that characterize surface water and groundwater interactions underwent linear regression and ANOVA statistical analyses to delineate any significant

relationships within and across fields. Linear regression models evaluated and compared interactions between total water applied ($TWA = IRR + PPT - RO$) and $DP$, $DP$ and GWLF, $TWA$ and GWLF. ANOVA analyses identified significant differences in means between the two study fields and different stations. Differences were considered significant at $\alpha = 0.05$. The 2020 irrigation season data collection only spanned mid-June through October (partial season), whereas the 2021 irrigation season data collection spanned April through October (complete season). Therefore, only 2021 data were included in the statistical analysis and presented in the Results section of this paper for optimal scientific consistency and comparability.

### 3. Results

*3.1. Irrigation Events and Deep Percolation Estimates*

The number of irrigation events and $DP$ varied between both fields over the 2021 irrigation season (Tables 2 and 3). Eight irrigation events took place on F1 (Table 2). A total of 24 irrigation events took place on F2 (Table 3). The average $IRR$ was 137 mm, and the $DP$ was 92 mm per irrigation event on F1 (Table 2). The average $IRR$ was 64 mm, and the $DP$ was 52 mm per irrigation event on F2 (Table 3). The F1 $DP$ estimates total was 739 mm, where 68% of the $IRR$ contributed to $DP$ (Table 4). For F2, the $DP$ estimates total was 1249 mm, where 81% of the $IRR$ contributed to $DP$ (Table 4).

**Table 2.** $DP$ results calculated with the water balance method for each irrigation event in the 2021 irrigation season for F1. This table shows the total time of irrigation, change in $\theta$ ($\Delta S$), total irrigation water applied ($IRR$), tailwater runoff ($RO$), total precipitation ($PPT$), and total $AET$ from the beginning of each irrigation event to 24 h after the end of irrigation. $DP$ estimates that resulted in negative values likely due to large $\Delta S$ values were considered to be 0, where no recharge occurred.

| Date | Irrigation Duration (h) | $\Delta S$ (mm) | $IRR$ (mm) | $RO$ (mm) | $PPT$ (mm) | $AET$ (mm) | $DP$ (mm) |
|---|---|---|---|---|---|---|---|
| 27 April 2021 | 49 | 55 | 158 | 0 | 0 | 8 | 94 |
| 4 May 2021 | 49 | 11 | 185 | 0 | 0 | 11 | 162 |
| 11 May 2021 | 48 | 105 | 197 | 2 | 0 | 14 | 76 |
| 18 May 2021 | 58 | −3 | 138 | 1 | 7 | 13 | 134 |
| 24 May 2021 | 45 | 23 | 235 | 47 | 0 | 18 | 147 |
| 1 June 2021 | 83 | −6 | 135 | 0 | 1 | 21 | 122 |
| 23 July 2021 | 70 | 18 | 12 | 0 | 22 | 11 | 5 |
| 31 July 2021 | 165 | 129 | 34 | 0 | 5 | 22 | 0 |
| Average | 71 | 42 | 137 | 6 | 4 | 15 | 92 |

**Table 3.** $DP$ results calculated with the water balance method for each irrigation event in the 2021 irrigation season for F2. This table shows the total time of irrigation, change in $\theta$ ($\Delta S$), total irrigation water applied ($IRR$), tailwater runoff ($RO$), total precipitation ($PPT$), and total $AET$ from the beginning of each irrigation event to 24 h after the end of irrigation. $DP$ estimates that resulted in negative values likely due to large $\Delta S$ values were considered to be 0, where no recharge occurred.

| Date | Irrigation Duration (h) | $\Delta S$ (mm) | $IRR$ (mm) | $RO$ (mm) | $PPT$ (mm) | $AET$ (mm) | $DP$ (mm) |
|---|---|---|---|---|---|---|---|
| 16 April 2021 | 70 | 46 | 313 | 0 | 8 | 6 | 270 |
| 10 May 2021 | 5 | −1 | 0 | 0 | 0 | 5 | 0 |
| 10 May 2021 | 8 | −1 | 5 | 0 | 0 | 1 | 6 |
| 11 May 2021 | 11 | 0 | 2 | 0 | 0 | 6 | 0 |
| 11 May 2021 | 11 | 0 | 2 | 0 | 0 | 0 | 2 |
| 12 May 2021 | 36 | 3 | 51 | 0 | 0 | 11 | 37 |
| 13 May 2021 | 9 | 0 | 2 | 0 | 0 | 0 | 2 |
| 14 May 2021 | 98 | 164 | 341 | 0 | 22 | 24 | 175 |
| 18 May 2021 | 10 | −55 | 1 | 0 | 4 | 2 | 68 |
| 19 May 2021 | 40 | −23 | 18 | 0 | 2 | 15 | 27 |

**Table 3.** *Cont.*

| Date | Irrigation Duration (h) | ΔS (mm) | IRR (mm) | RO (mm) | PPT (mm) | AET (mm) | DP (mm) |
|---|---|---|---|---|---|---|---|
| 23 May 2021 | 2 | −3 | 0 | 0 | 0 | 2 | 1 |
| 24 May 2021 | 1 | −2 | 1 | 0 | 0 | 4 | 0 |
| 24 May 2021 | 18 | −6 | 0 | 0 | 0 | 10 | 0 |
| 29 May 2021 | 33 | −5 | 8 | 0 | 0 | 7 | 5 |
| 31 May 2021 | 23 | −4 | 3 | 0 | 16 | 7 | 15 |
| 5 June 2021 | 78 | −5 | 216 | 3 | 2 | 31 | 189 |
| 26 June 2021 | 112 | 82 | 397 | 8 | 11 | 10 | 308 |
| 1 August 2021 | 58 | 6 | 26 | 0 | 7 | 12 | 15 |
| 18 August 2021 | 7 | −3 | 1 | 0 | 0 | 3 | 1 |
| 18 August 2021 | 39 | −4 | 39 | 0 | 0 | 6 | 37 |
| 27 August 2021 | 8 | −4 | 12 | 0 | 0 | 6 | 10 |
| 29 August 2021 | 18 | −4 | 20 | 0 | 0 | 7 | 17 |
| 8 September 2021 | 50 | −4 | 70 | 0 | 0 | 11 | 63 |
| 11 September 2021 | 8 | −1 | 2 | 0 | 0 | 6 | 0 |
| Average | 31 | 7 | 64 | 0 | 3 | 8 | 52 |

**Table 4.** Summary table displaying total number of irrigation events, cumulative *IRR*, *DP*, and percent of *IRR* that contributed to *DP* for each field over the 2021 irrigation season.

| Field | Year | Number of Irrigation Events | IRR (mm) | DP (mm) | Percent DP (%) |
|---|---|---|---|---|---|
| F1 | 2021 | 8 | 1093 | 739 | 67.7 |
| F2 | 2021 | 24 | 1541 | 1249 | 81.1 |

While annual variability is common due to differing environmental conditions, surface water availability, and irrigation scheduling, monthly irrigation summaries and averages on both fields demonstrate similar ranges of water balance parameters between 2020 and 2021 (Table 5). On F1 in 2020, *DP* averaged 12 mm and 29 mm per irrigation event in July and August, respectively, with no irrigations in September. In 2021, the average *DP* for July was 2 mm with no irrigations in August or September. No irrigations took place on F2 in July in 2020 and 2021. In 2020, F2 *DP* averaged 9 mm in August and 1 mm in September. In 2021, *DP* averaged 16 mm in August and 32 mm in September.

**Table 5.** Comparison of monthly number of irrigation events, average ΔS, average *IRR*, average *RO*, average *PPT*, average *DP*, and total *DP* for F1 and F2 for three months in 2020 and 2021. The months chosen for comparison are July through September because these were the first complete monthly records after data collection began in 2020 (data collection began early-June 2020) to ensure optimal comparability between the two irrigation seasons on both fields.

| Field | Month | Number of Irrigation Events | Avg ΔS (mm) | Avg IRR (mm) | Avg RO (mm) | Avg PPT (mm) | Avg AET (mm) | Avg DP (mm) | Sum DP (mm) |
|---|---|---|---|---|---|---|---|---|---|
| F1 | July 2020 | 2 | −3 | 13 | 0 | 1 | 4 | 12 | 24 |
| | August 2020 | 1 | −3 | 31 | 0 | 5 | 10 | 29 | 29 |
| | September 2020 | 0 | | | (no irrigation events) | | | | |
| | July 2021 | 2 | 73 | 23 | 0 | 13 | 16 | 2 | 5 |
| | August 2021 | 0 | | | (no irrigation events) | | | | |
| | September 2021 | 0 | | | (no irrigation events) | | | | |

**Table 5.** *Cont.*

| Field | Month | Number of Irrigation Events | Avg ΔS (mm) | Avg *IRR* (mm) | Avg *RO* (mm) | Avg *PPT* (mm) | Avg *AET* (mm) | Avg *DP* (mm) | Sum *DP* (mm) |
|---|---|---|---|---|---|---|---|---|---|
| F2 | July 2020 | 0 | | | (no irrigation events) | | | | |
| | August 2020 | 3 | 2 | 11 | 0 | 5 | 7 | 9 | 27 |
| | September 2020 | 2 | −1 | 6 | 0 | 0 | 6 | 1 | 3 |
| | July 2021 | 0 | | | (no irrigation events) | | | | |
| | August 2021 | 5 | −2 | 19 | 0 | 1 | 7 | 16 | 80 |
| | September 2021 | 2 | −3 | 36 | 0 | 0 | 8 | 32 | 63 |

The linear regression analysis showed a positive, significant relationship between *DP* and *TWA* and *TWA*-Δ*S* for both F1 ($p = 8.37 \times 10^{-3}$ and $p = 3.48 \times 10^{-4}$, respectively) and F2 ($p = 7.88 \times 10^{-16}$ and $p < 2.00 \times 10^{-16}$, respectively) (Table 6). Previous research in the region found that prior $\theta$ significantly impacted *DP* [50], which is why we included *TWA*-Δ*S* in the linear regression. Additionally, F2 exhibited a significant positive relationship between *DP* and irrigation duration ($p = 5.42 \times 10^{-8}$). ANOVA showed statistically significant differences in the mean irrigation duration and mean number of irrigation events between F1 and F2 (Table 7).

**Table 6.** Statistics from linear regression models comparing *DP* and *TWA, DP* and *TWA*-Δ*S*, and *DP* and irrigation duration for irrigation events on each field in 2021. Significant relationships are highlighted by *p* values with an asterisk (*).

| Field | *t* | $R^2$ | *p* |
|---|---|---|---|
| | TWA (mm) | | |
| F1 | 3.86 | 0.713 | $8.37 \times 10^{-3}$ * |
| F2 | 16.4 | 0.925 | $7.88 \times 10^{-16}$ * |
| | TWA-ΔS (mm) | | |
| F1 | 7.26 | 0.898 | $3.48 \times 10^{-4}$ * |
| F2 | 67.9 | 0.995 | $<2.00 \times 10^{-16}$ * |
| | Irrigation duration (h) | | |
| F1 | −2.12 | 0.427 | 0.0787 |
| F2 | 8.04 | 0.746 | $5.42 \times 10^{-8}$ * |

**Table 7.** ANOVA tests conducted with the field as the independent variable and different variables of interest as dependent variables to identify significant differences in means between surface water and groundwater interactions and irrigation management across both fields for 2021 irrigation events. Significant differences are highlighted by *p* values with an asterisk (*).

| Dependent Variable | *F* | $R^2$ | *p* |
|---|---|---|---|
| DP (mm) | 1.40 | 0.0447 | 0.245 |
| TWA (mm) | 2.23 | 0.0691 | 0.146 |
| TWA-ΔS (mm) | 0.811 | 0.0263 | 0.375 |
| Irrigation duration (h) | 8.17 | 0.214 | $7.67 \times 10^{-3}$ * |
| Number of irrigation events | 9.66 | 0.244 | $4.09 \times 10^{-3}$ * |

*3.2. Shallow Groundwater Response to Irrigation Inputs*

GWLF and response to irrigation inputs were observed for both study fields over the 2020 and 2021 irrigation seasons (Figure 2). In 2021 on F1, gw1 GWLF averaged 533 mm, gw2 GWLF averaged 262 mm, and gw3 GWLF averaged 863 mm. The greatest observed GWLF of the F1 monitoring wells was 1699 mm on 11 May 2021 (Table 8).

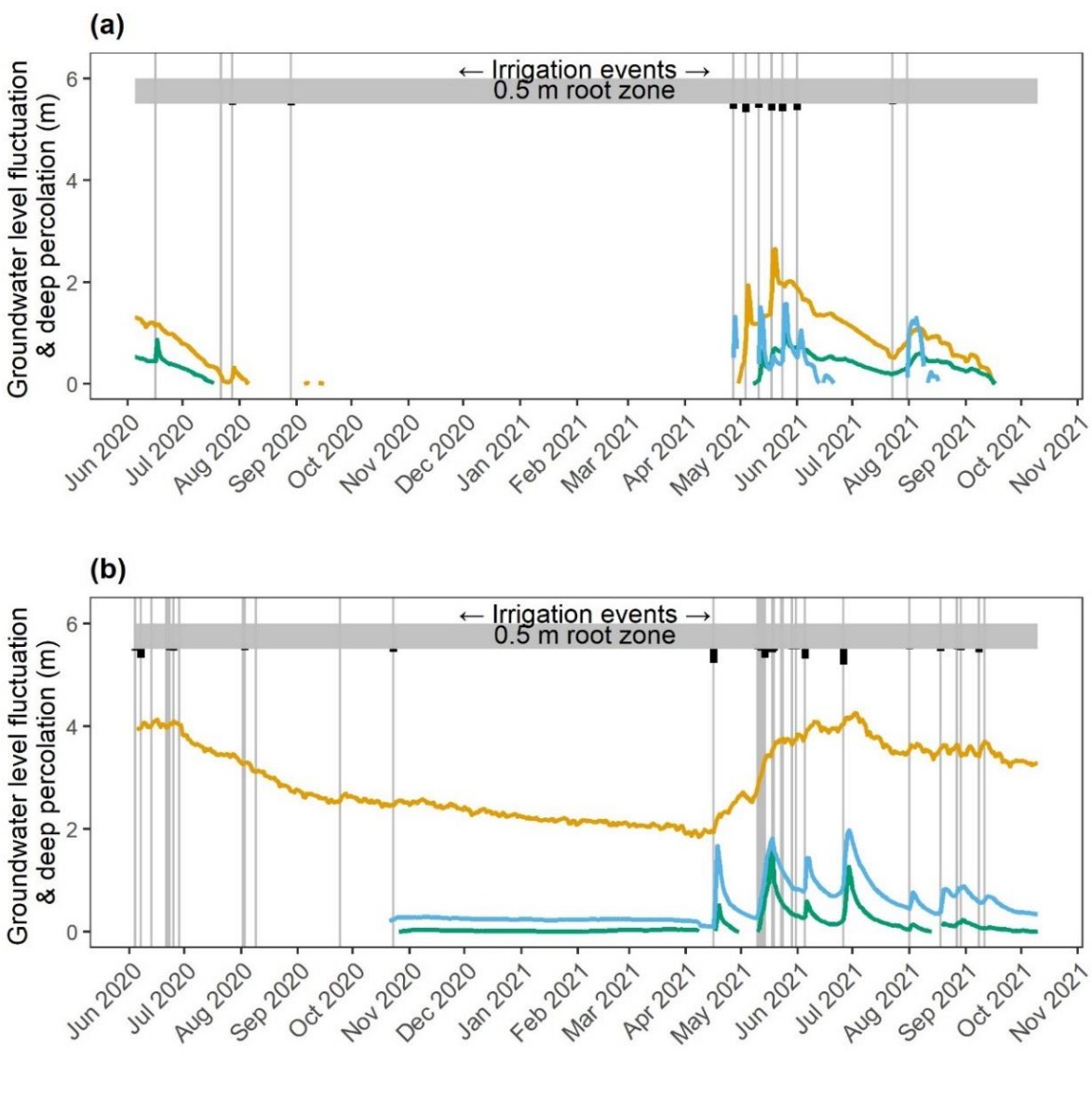

**Figure 2.** Shallow groundwater levels for all monitoring wells, irrigation events (vertical gray lines), and DP estimates for the study sites: (**a**) F1; (**b**) F2 from early June 2020 through October 2021.

**Table 8.** GWLF (mm) in response to irrigation events in 2021 for all wells on F1 calculated as the difference between groundwater level prior to the irrigation onset (averaged over the 6 h prior to the irrigation onset) and maximum water level rise until the following irrigation event.

| Date | GWLF gw1 (mm) | GWLF gw2 (mm) | GWLF gw3 (mm) |
|---|---|---|---|
| 27 April 2021 | 452 | 0 | 1549 |
| 4 May 2021 | 1601 | 55 | 0 |
| 11 May 2021 | 197 | 781 | 1699 |
| 18 May 2021 | 1363 | 280 | 288 |
| 24 May 2021 | 55 | 428 | 1350 |
| 1 June 2021 | −6 | 108 | 618 |
| 23 July 2021 | 320 | 150 | 114 |
| 31 July 2021 | 282 | 290 | 1282 |
| Average | 533 | 262 | 863 |

Both study fields exhibited sharp groundwater response to irrigation events and *DP* (Figure 2). F1 groundwater levels would generally show a moderate decline after the peak GWLF (Figure 2a). F1 gw1, located next to the irrigation canal, maintained more elevated groundwater levels for longer than the other two wells on this field. Gw3 displayed the "flashiest" response to irrigation events and *DP* both in the rise and fall around the peak GWLF.

F2 groundwater levels—specifically gw2 and gw3—would decline more rapidly following the peak GWLF (Figure 2b). F2 gw1 showed more short-term fluctuation due to pumping water for residences on the property and more gradual rise and fall to the beginning and end of the irrigation season due to its deeper reach, upgradient position, and closer tie to ditch seepage from nearby acequias and water delivery canals rather than irrigation events. On F2, GWLF in gw1 differed from the groundwater response in the other monitoring wells to irrigation events (Figure 2b). This well (gw1) was the landowner's drinking water well that was 13 m deep and located upgradient of the irrigated field (Figure 1). GWLF from F2 gw1 levels were likely related to acequia flow as opposed to irrigation events. The main acequia flowed along the south border of the property, and the intermediate ditch that delivered water from the acequia onto F2 flowed next to gw1, so ditch seepage from delivery canals likely supplied this well. F2 gw1 GWLF averaged 167 mm in 2021. For the other F2 monitoring wells gw2 and gw3, GWLF averaged 210 mm and 272 mm, respectively. The greatest observed GWLF of the F2 monitoring wells was 1697 mm on 14 May 2021 (Table 9).

**Table 9.** GWLF (mm) in response to irrigation events in 2021 for all wells on F2 calculated as the difference between groundwater level prior to the irrigation onset (averaged over the 6 h prior to the irrigation onset) and maximum water level rise until the following irrigation event.

| Date | GWLF gw1 (mm) | GWLF gw2 (mm) | GWLF gw3 (mm) |
|---|---|---|---|
| 16 April 2021 | 876 | 602 | 1677 |
| 10 May 2021 | 100 | 7 | 42 |
| 10 May 2021 | 87 | 21 | 87 |
| 11 May 2021 | 76 | 141 | 129 |
| 11 May 2021 | 30 | 132 | 103 |
| 12May 2021 | 278 | 275 | 288 |
| 13 May 2021 | 53 | −3 | 20 |
| 14 May 2021 | 242 | 1697 | 969 |
| 18 May 2021 | 3 | −45 | −23 |
| 19 May 2021 | 254 | −40 | 47 |
| 23 May 2021 | 40 | −7 | −9 |
| 24 May 2021 | −35 | −8 | −15 |
| 24 May 2021 | 57 | 3 | 1 |
| 29 May 2021 | 198 | −1 | −4 |
| 31 May 2021 | 164 | 30 | 33 |
| 5 June 2021 | 362 | 418 | 795 |
| 26 June 2021 | 247 | 1179 | 1204 |
| 1 August 2021 | 93 | 140 | 345 |
| 18 August 2021 | 41 | 0 | 43 |
| 18 August 2021 | 168 | 220 | 430 |
| 27 August 2021 | 30 | 96 | 147 |
| 29 August 2021 | 250 | 145 | 76 |
| 8 September 2021 | 368 | 3 | 28 |
| 11 September 2021 | 33 | 33 | 106 |
| Average | 167 | 210 | 272 |

Linear regression statistical analysis identified any significant relationships between *DP* and GWLF of each monitoring well as well as *TWA* and GWLF of each monitoring well for both study fields (Table 10). No significant relationships were identified between GWLF and *DP* nor GWLF and *TWA* for any F1 monitoring wells. This is likely related to the

land manager's use of several headgates spread out along the southern field border used at different times unevenly applying irrigation water. All wells on F2 exhibited positive significant relationships between GWLF and *DP*, and GWLF and *TWA* (Table 10). However, F2 gw1 GWLF was likely related to acequia flow and ditch seepage rather than irrigation events due to its upgradient position.

**Table 10.** Statistics from linear regression models comparing GWLF and *DP* as well as GWLF and *TWA* for all monitoring wells on each field from data collected over the 2021 irrigation season. Significant relationships are highlighted by *p* values with an asterisk (*).

| | GWLF gw1 (mm) | | | GWLF gw2 (mm) | | | GWLF gw3 (mm) | | |
|---|---|---|---|---|---|---|---|---|---|
| | *t* | $R^2$ | *p* | *t* | $R^2$ | *p* | *t* | $R^2$ | *p* |
| | | | | DP (mm) | | | | | |
| F1 | 1.19 | 0.190 | 0.280 | −0.252 | 0.0105 | 0.810 | −0.426 | 0.0294 | 0.685 |
| F2 | 4.60 | 0.490 | $1.40 \times 10^{-4}$ * | 5.78 | 0.603 | $8.09 \times 10^{-6}$ * | 11.5 | 0.858 | $8.77 \times 10^{-11}$ * |
| | | | | TWA (mm) | | | | | |
| F1 | 0.540 | 0.0464 | 0.608 | 0.720 | 0.0795 | 0.499 | 0.639 | 0.0638 | 0.546 |
| F2 | 3.93 | 0.413 | $7.08 \times 10^{-4}$ * | 10.3 | 0.828 | $7.37 \times 10^{-10}$ * | 11.4 | 0.856 | $9.91 \times 10^{-11}$ * |

Figure 3 provides a visualization of how the groundwater level data from the three monitoring wells compare across the two study fields. The greatest variation is apparent between gw1 on F1 and F2 because the well on F2 is a residential drinking well and is much deeper (see Section 2.2.5 for more detailed metrics regarding the groundwater monitoring wells included in this study).

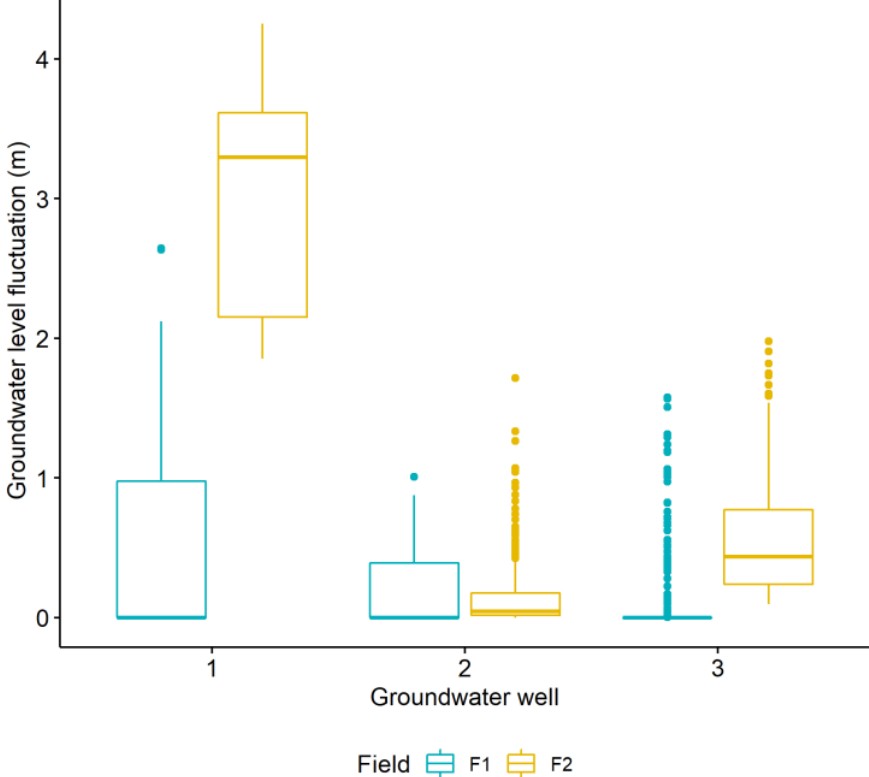

**Figure 3.** Boxplots created using daily averages of GWLF data from April 2021 through October 2021 visually comparing the medians, quartiles, and ranges of the three monitoring wells across both study fields.

## 4. Discussion

Our results showed both fields have significant relationships between *DP* and *TWA* and between *DP* and *TWA-ΔS* (Table 6). One field, F2, exhibited significant relationships between *DP* and irrigation duration (Table 6), GWLF and *DP*, and GWLF and *TWA* (Table 10). Antecedent soil moisture and soil conditions are particularly important factors to consider when discussing *DP*. More irrigation water is needed to saturate the ERZ when antecedent soil moisture is low or at times of greater plant water use which results in potentially less groundwater recharge from a given amount of irrigation water applied. *DP* was not significantly different when compared across both fields. The only significant differences when comparing irrigation events and *DP* estimates across both fields were irrigation duration and the number of irrigation events (Table 7). These results indicate that surface water and groundwater are tightly connected in this area, but variation in *DP* and groundwater response exists between land managers and fields due to differing irrigation practices.

Although we only report data and results from the 2021 irrigation season, our data collection began in June 2020. When comparing monthly averages and totals for months with complete records for both 2020 and 2021, several patterns emerge regarding irrigation scheduling, average ΔS, and average *DP* (Table 5). F1 irrigation frequency tapered off toward the end of the irrigation season for both 2020 and 2021, while F2 irrigation frequency increased toward the end of the season. Average ΔS was constant between the two fields over both years of data collection, ranging from −3 to 2 mm with a notably large value for F1 in July 2021 (73 mm) as an outlier perhaps related to frequent rainfall that occurred around that time of year and uneven irrigation water application. On F1, the average *DP* ranged from 2 to 29 mm over 2020 and 2021. Similarly, the F2 average *DP* ranged from 1 to 32 mm. These patterns help validate our water balance results (Tables 2 and 3) by demonstrating consistent and comparable water balance components and *DP* estimates across both fields over 2020 and 2021.

Previous acequia research in northern New Mexico forage fields that also used water balance methodology to estimate *DP* reflects similar results (Table 11), illustrating that we appropriately captured acequia surface water and groundwater interactions and irrigation practices. Our results reflected the greatest *DP* season totals (739 and 1249 mm) and the greatest percentage of *IRR* that contributed to *DP* (68 and 81%), critically filling in the range of possible seasonal *DP* values and characteristics by refining our understanding of acequia irrigation-related recharge in the context of long-term field data collection in northern New Mexico.

**Table 11.** A comparison of how our *DP* estimates compare to similar studies that used a water balance approach to estimate *DP* in forage grass fields in northern New Mexico.

| Study | Location & Year | Average DP per Irrigation Event (mm) | Sum DP over Irrigation Season (mm) | Percent DP over Irrigation Season (%) |
|---|---|---|---|---|
| Ochoa et al. (2013) | Alcalde 2005 | 107 | 533 | 46 |
| Ochoa et al. (2013) | Alcalde 2006 | 119 | 476 | 48 |
| Gutiérrez-Jurado et al. (2017) | Rio Hondo (F1) 2013 | 53 | 531 | 51 |
| Gutiérrez-Jurado et al. (2017) | Alcalde 2013 | 55 | 382 | 39 |
| Gutiérrez-Jurado et al. (2017) | El Rito 2013 | 77 | 462 | 31 |
| Conrad et al. (this paper) | Rio Hondo (F1) 2021 | 92 | 739 | 68 |
| | Rio Hondo (F2) 2021 | 52 | 1249 | 81 |

Observations and projections of changing climate and snowmelt dynamics within the Rio Grande Basin—specifically the Upper Rio Grande headwaters region—are of particular interest to researchers and stakeholders due to the reliance of downstream users on snowmelt-dominated subbasins to meet water availability needs. For example, streamflow at Fort Quitman, Texas, USA has decreased 95% relative to the river's native streamflow [51]. In the Colorado River Basin, temperature-driven "hot droughts" have

been connected to increased sublimation of snow which results in less runoff from a given snowpack [52]. Similarly, the interannual variability of streamflow related to snow water equivalent (SWE) has decreased by 40% in the Upper Rio Grande Basin, indicating that the connection between peak SWE and runoff volume is substantially weaker [7]. This drift between SWE and runoff is particularly critical because a large portion—50 to 75%—of the Rio Grande streamflow is sustained by seasonal snowpack accumulation [53]. Through paleoclimate reconstructions published in 2017, researchers identified a 30-year declining trend in runoff ratio since the 1980s which appeared unprecedented in the context of the last 440 years [54]. Observed, historical mean winter and spring temperatures have significantly increased in the Upper Rio Grande Basin [7], and temperatures rose at an alarming rate of 0.4 °C per decade from 1971 through 2011, informing temperature predictions of a 2 to 3 °C increase in average temperature by the end of the 21st century [8]. The SWE on April 1 has significantly decreased by 25% [7], where the mean melt season snow covered area is predicted to decrease 57 to 82%, and peak flow is predicted to arrive 14 to 24 days earlier than usual [6].

The combination of increasing temperature and more variable precipitation inputs are expected to create a decrease in summertime flows and increase the frequency, intensity, and duration of floods and droughts in the Upper Rio Grande Basin [8]. Elias et al. (2015) found that total annual runoff volume of Upper Rio Grande subbasins and tributaries could increase 7% in wetter scenarios but decrease 18% in drier scenarios. In the Rio Hondo watershed, annual daily mean streamflow has significantly decreased 0.85% per year since water year 1976 [55]. Another study found that the Rio Hondo baseflow, runoff, and streamflow have also significantly decreased since water year 1980 due to decreasing snowmelt rates [56].

Decreasing surface water flows in the Upper Rio Grande region will have negative effects on acequia water availability for acequia communities in this region. A previous study conducted in the Rio Hondo Valley found statistically significant relationships between river and acequia flows [57]. Similarly, spatial analysis of the normalized difference vegetation index (NDVI) found that the irrigated landscape within the Rio Hondo Valley expanded and contracted in response to wet or dry years, showing that irrigation intensity varied with available surface water [58,59]. Therefore, in the Rio Hondo Valley, acequia flow is directly related to river flow, and the variability of acequia irrigation intensity is apparent in wet and dry years. As a result, the irrigated landscape and acequia irrigation decrease as surface water resources decrease.

If surface water river flows continue to decrease, then acequia water availability and the acequia-irrigated landscape will decrease, as will regular *DP* and groundwater recharge [60]. As a mechanism that temporarily stores surface water in the subsurface which eventually returns to the river system as delayed return flow, *DP* can serve as a very important buffer against climate change; however, mean recharge in Taos County first significantly decreased in 1996 [61]. Baseflow is also an extremely critical element of the hydrologic regime in the Upper Rio Grande Basin where baseflow contributions account for 49% of total discharge upstream of Albuquerque, New Mexico [56]. Surface water and groundwater connectivity is critical for continued baseflows, and acequia-related *DP* and return flows play an important role in maintaining this connection. As climate change continues to negatively impact surface water availability and groundwater recharge in northern New Mexico and the Upper Rio Grande Basin, both acequia communities and the state of New Mexico will have to decide how to adapt to new climatic and hydrologic regimes.

When considering water use and management practices, either as a water manager or for modeling purposes, it is critical to determine the type and direction of adaptation (e.g., adaptation or maladaptation) occurring in response to climate change stressors [62]. Maladaptive actions are enacted to prevent or reduce vulnerability associated with climate change but ultimately have adverse impacts or increase vulnerabilities in the same or related systems. Examples of adverse impacts include: (1) an increase greenhouse gas emissions; (2) a disproportionate impact on vulnerable populations; (3) high environmental

opportunity costs; (4) reduced incentives to adapt; and (5) dependencies that limit future generations [63]. Unfortunately, all too often, water management adaptation and governance strategies are maladaptive, such as water operations in Flint, Michigan [64], water deliveries in California's San Joaquin Valley [65], and development in Australian coastal cities [66].

Adaptive management practices are more prepared for climate change by incorporating flexibility and responsiveness into water management institutions and governance structures [67–72]. While some suggest doing this through intraregional contracts and mergers [67], acequias have been doing this for centuries through a concept known as *repartimiento*—the ability to employ flexible and dynamic water deliveries to distribute water as equitably as possible by sharing water shortages either within a single acequia or between different acequias throughout a given watershed. Cruz et al. (2019) documented this phenomenon by showing that the water available in acequias is directly correlated to the water available in the stream system [57]. When not enough surface water is available to irrigate crops, landowners will typically irrigate a smaller parcel of their total crop land as opposed to the entire area. This shows the inherent adaptability embedded within traditional acequia irrigation frameworks that is and will continue to be crucial in the context of a changing climate, growing seasons, and streamflow regimes.

The flood irrigation regime these two fields and the greater Rio Hondo Valley—as well as other acequia communities—follow experience groundwater recharge benefits inadvertently associated with managed aquifer recharge (MAR). Recently, many research articles [73] have featured different MAR techniques and pilot programs [34,74]. MAR is an approach used to replenish groundwater resources and is becoming more common in areas of heavy groundwater pumping and declining aquifer levels. There are many different techniques and objectives within this overarching mitigation approach. One promising approach that utilizes already existing infrastructure is applying MAR to irrigated agricultural lands, where surface water is applied over large areas as opposed to the more traditional MAR approach of facilitating high recharge at dedicated recharge sites [34]. This form of MAR reduces costs associated with infrastructure, piping, and energy given the gravity-driven water distribution [75]. This framework is naturally mirrored on a regional scale in northern New Mexico's acequia networks. Acequia networks divert surface water through a system of (typically earthen) canals to fields for flood irrigation, where seepage occurs throughout time in the canals and application in the fields. Acequia irrigators greatly value these contributions to groundwater for the many environmental and water storage benefits the recharge provides (Figure 4). Acequias are not without their challenges, but they can serve as a model for sustainable, integrated water management that implicitly employs MAR and welcomes groundwater recharge as a benefit rather than an inefficiency [76].

Characterized by regular shallow aquifer recharge and flexible and dynamic water management that reflects equity and current environmental conditions, acequias offer several reasons why we should consider maintaining traditional irrigation systems in the face of climate change (Figure 5). In times of reduced surface water availability, acequia irrigators only irrigate smaller parcels of their total irrigated land and typically invest in deep rooted, drought-tolerant crops that can persist through growing seasons without much irrigation water application. Acequia communities have followed this model traditional flood irrigation model and persisted through drought for hundreds of years in northern New Mexico. However, when thinking about the future, the question then becomes: How should acequia communities adapt to meet reduced surface water availability and changing streamflow regime challenges that the current prolonged and unprecedented drought presents if traditional acequia irrigation practices are no longer sufficient?

Traditional acequia operations are typically associated with resilience [77], but many acequia irrigators and managers are unsure of how sustainable certain adaptations are moving forward (e.g., lining earthen irrigation canals, switching from flood to drip irrigation, greater reliance on groundwater pumping) and the implications of any detrimental effects on groundwater levels (i.e., lowering the water table) which are ultimately connected to

surface water availability (Figure 5). Lining irrigation ditches, switching from flood to drip irrigation, and supplemental groundwater pumping are commonly called into question by acequia community members which is why these adaptation strategies are highlighted in this paper. While these three strategies can be beneficial, lining ditches and drip irrigation reduce pathways for surface water to seep into the groundwater, and a growing reliance on groundwater pumping will negatively impact surface water and groundwater connectivity by lowering the water table (Figure 5). When used simultaneously in a region where baseflow is a crucial component of sustaining Rio Grande streamflow [56] and traditional acequia irrigation related recharge serves as delayed return flow [22], the reduction in groundwater recharge and the increase in groundwater pumping will negatively impact surface water availability for downstream users and begin propagating a cycle of maladaptation. It will be critical to prioritize traditional flood irrigation approaches and benefits such as groundwater recharge as much as possible when acequia communities or similar community-based irrigation systems are seeking solutions under conditions of reduced surface water availability.

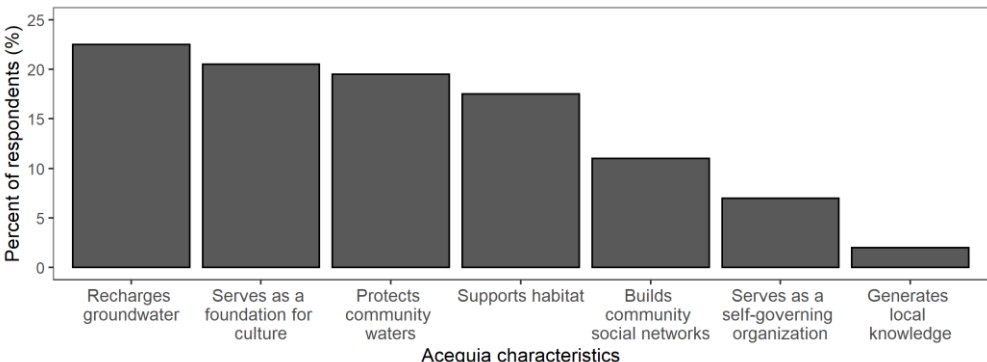

**Figure 4.** When posed the question: "In addition to providing irrigation water for local uses, which of the following characteristics of acequias are most important to you?", Rio Hondo Valley acequia community members (22.5%) reported groundwater recharge as the most valued attribute out of a variety of environmental, social, cultural, and governance options (*n* = 25). These data were collected as background information from adaptive capacity pre- and post-survey instruments distributed to the Rio Hondo acequia community. The final percent of respondents were averaged across the two surveys [55].

It is important to distinguish between modernization of irrigation *infrastructure* and modernization of irrigation *management*. While lining ditches, switching to drip, and supplemental groundwater pumping focus on using surface water more efficiently through engineering and infrastructure improvements, water managers and irrigators must be provided with tools, resources, and information that enable efficient and adaptive water management and allocation. One example of this is real-time monitoring accessible through a web interface which has been shown to increase adaptive capacity indicators within the Rio Hondo acequia community [55]. With water scarcity only becoming a more pressing issue in the Southwest within the context of climate change, it will be critical to continue evaluating the adaptability of water management and agricultural production approaches, reflect findings in new and transformative policy, and ask ourselves if we should be modernizing infrastructure or management to avoid falling into the irrigation efficiency paradox trap [78,79].

A key element for the success of acequia and other community-based irrigation systems is community water management system functionality (see Figure 5). To have a functioning community water management system, there must first be a community to manage and use the water, so individuals must see value in acequias or acequia irrigation. When researchers explored capital gained within acequia communities, they found that only about 30% of family income was generated from acequia agriculture and that

external income helped sustain acequia-irrigated properties and agriculture [80]. Surveys and interviews revealed that connection to land, water, and community were the values that drove acequia community members to respond to adverse circumstances (e.g., economic hardship, population growth, drought, increased development), demonstrating that acequia communities are founded and fueled by values within the *moral* economy rather than the typical market economy [30,59,80]. Therefore, identifying appropriate irrigation modernization recommendations must consider irrigation community motivations or values and be tailored toward enabling water management system functionality. While acequias foster long-term resilience, short-term vulnerabilities that impact acequia irrigation are surface water shortages. More work is needed to assess the specific impacts of changing irrigation regimes and technologies in acequia regions and identify adaptations that optimize groundwater recharge while also taking declining surface water availability into consideration.

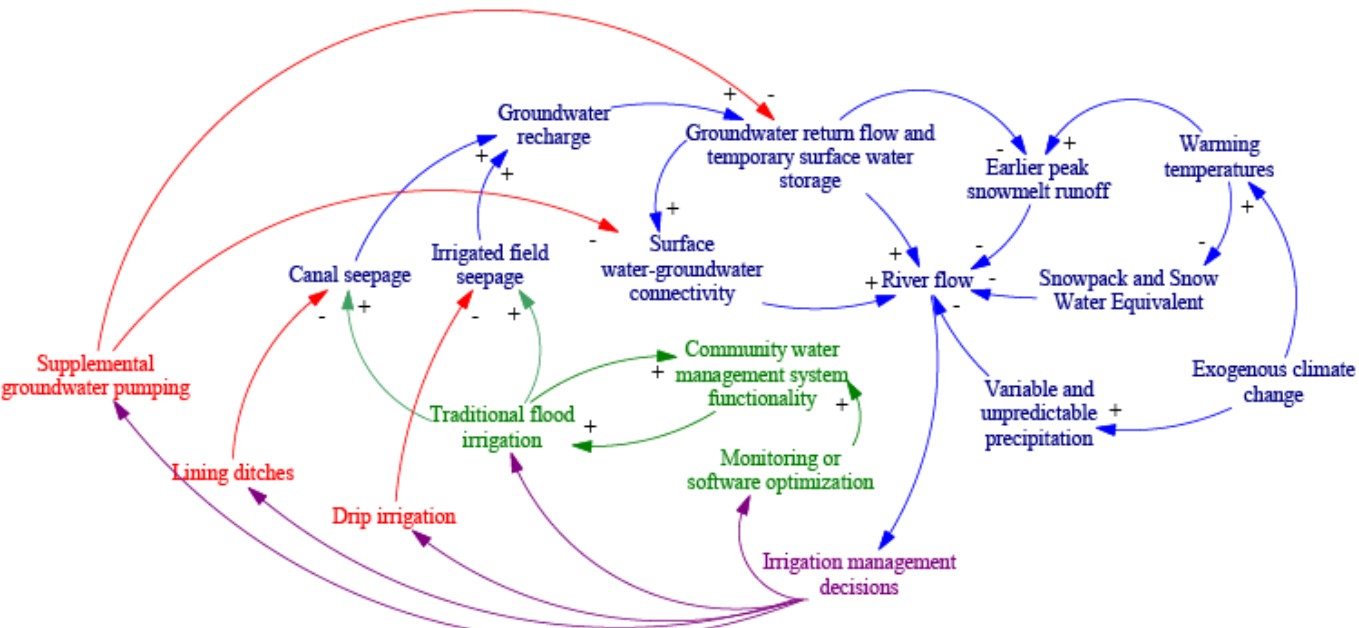

**Figure 5.** Causal loop diagram (CLD) showing the interactions and connections between environmental phenomena (blue), decision making (purple), adaptive management (green), and potentially maladaptive management (red) of acequia irrigation systems where traditional flood irrigation is assumed to only use surface water. Please note that the potentially maladaptive management options might be considered adaptive for other regions and irrigation regimes outside the scope of this paper.

## 5. Conclusions

In this study, we compared surface water and groundwater interactions and shallow aquifer response to irrigation events in two flood-irrigated forage grass fields located within the same acequia-irrigated valley in northern New Mexico, USA. Our results indicate that surface water and groundwater are tightly connected in this area, but variations in *DP* and groundwater response exists between land managers and fields due to differing flood irrigation scheduling and management. Additionally, while our results are consistent with previous water balance studies conducted in acequia-irrigated forage grass fields in northern New Mexico, this is the first paper to relate the findings from all the similar studies in the region since the first study was conducted in 2005. Because recharge acequia irrigation-related recharge eventually becomes delayed return flow to rivers [22], studies such as this are critical for determining how surface water and groundwater connectivity changes over time as it directly impacts surface water availability for downstream users [56]. We expect less *DP* will occur in acequia-irrigated fields if future climate change predictions and warming trends continue.

The *DP* and shallow groundwater recharge that occur as a byproduct of acequia flood irrigation are the exact management benefits needed for appropriate climate change adaptation. By maintaining recurring and consistent groundwater recharge, stream systems stay watered, which enables valley and regional cooperation between acequia-governing systems to continue. Alternatively, if acequia regions begin relying more heavily on groundwater pumping (for example), water tables would drop, making less water available in stream systems as surface water and groundwater become disconnected. These actions would propagate a cycle of maladaptation by undermining the hydrologic functions and community collaboration that make acequias so sustainable.

**Author Contributions:** Conceptualization, L.M.C., A.G.F. and C.G.O.; methodology, L.M.C., A.G.F. and C.G.O.; validation, L.M.C., A.G.F., S.J.G. and C.G.O.; formal analysis, L.M.C.; investigation, L.M.C., A.G.F. and C.G.O.; resources, L.M.C., A.G.F. and C.G.O.; data curation, L.M.C.; writing—original draft preparation, L.M.C.; writing—review and editing, L.M.C., A.G.F., S.J.G. and C.G.O.; visualization, L.M.C.; supervision, A.G.F. and C.G.O.; project administration, L.M.C. and A.G.F.; funding acquisition, A.G.F. All authors have read and agreed to the published version of the manuscript.

**Funding:** This research was funded by the state of New Mexico, through special and state appropriations made to the New Mexico Water Resources Research Institute NMWRRI2019-2020 and NMWRRI2021, and New Mexico State University's College of Agricultural, Consumer and Environmental Sciences.

**Institutional Review Board Statement:** Not applicable.

**Informed Consent Statement:** Not applicable.

**Data Availability Statement:** The data and statistical analysis presented in this study are openly available within the New Mexico Water Resources Research Institute's Data Set Repository at the following link: https://nmwrri.nmsu.edu/ds-002/ (accessed on 8 March 2022).

**Acknowledgments:** We would like to sincerely thank local stakeholders for their involvement, support, collaboration, and for their interest in understanding how irrigation practices relate to surface water and groundwater interactions.

**Conflicts of Interest:** The authors declare no conflict of interest. The funders had no role in the design of the study; in the collection, analyses, or interpretation of data; in the writing of the manuscript, or in the decision to publish the results.

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
