# Peer review of "A Water Balancing Act: Water Balances Highlight the Benefits of Community-Based Adaptive Management in Northern New Mexico, USA"

_hydrology, doi:10.3390/hydrology9040064_

Round 1
Reviewer 1 Report
Thank you for opportunity to review your paper - it is well written and addresses issues that will be of increasing focus in coming years. I have made one minor suggestion to improve Figure 5 (add a definition for SWE as it doesn't appear anywhere in the paper).
Decisions about irrigation modernization are often focused on physical interventions to improve water delivery and application efficiency without adequate recognition of the broader hydrological implications of such changes on the overall water environment within the river basin. You present a useful approach to assessing potential adverse impacts of modernization to improve local irrigation performance.
I think it would have been useful to provide some indication of the level of water stress across the basin - given your observation that climate change is driving declining snow-pack and thus availability and timing of water resource availability. Managed aquifer recharge may provide a mechanism to (partially?) substitute groundwater storage for the reduced storage in the snowpack - but it is not clear to me that the continued traditional management of the acequia system would result in a sufficient adaptation to the changing water availability resulting from climate change. Perhaps you could include something about the potential requirement to "modernize the management" of the acequia systems to compensate for changing water availability - i.e. a software change rather than a change in the lining or application methods as posited in Fig 5.
Although I have no experience of acequia systems in USA I have worked on similar cascading and informally managed irrigation systems in Asia. Where interventions to modernize these systems without adequate assessment and understanding of the interconnection of the various elements of the hydrological system and the existing adaptations made by the existing management practices have led to sub-optimal, or even, adverse irrigation outcomes. An approach similar to yours may have prevented to perverse and disappointing investments.
I am not sure your title is fully appropriate - perhaps not "Is Irrigation Modernization the Answer?" but rather "Identifying locally appropriate irrigation modernization?" However, your more provocative title certainly catches the eye - so please do keep it.

Reviewer 2 Report
The manuscript ‘Is Irrigation Modernization the Answer? Surface Water and Groundwater Interactions Justify Community-Based Adaptive Management in an Irrigated Valley in Northern New Mexico, USA’ treats the interesting subject of surface water and groundwater interactions in the framework of community-based irrigation water management.
The authors analyze the water balance in two fields in the common water delivery system for flood irrigation in New Mexico with acequia networks which face many socio-environmental challenges.
The article is well structured, well documented, and well written.
Improvement of the manuscript is necessary to be published in the Hydrology MDPI Journal.
The authors should take under consideration the following remark:
More comments are needed concerning the conditions, under which the proposed system is adequate. What should be the solution in case the water resources are very reduced? There are only few comments at the end of the manuscript (lines 593-597). Please develop more the adequate solution under these conditions.
Reviewer 3 Report
Quantifying groundwater recharge from irrigation in water-scarce regions is critical for sustainable water management. The authors provide an interesting case study in estimating groundwater recharge. However, the primary shortage of this paper is the lack of innovation. The two hypotheses “1) DP and total water applied, and DP and groundwater response would be positively related on both fields and 2) DP and total water applied would be significantly different across both study fields” are becoming common knowledge in this research field. Moreover, considering the shallower water table depth, surface water and groundwater are naturally connected, without additional evidence.
Besides, the title and abstract are not the same thing. Therefore, I recommend rejecting this manuscript.
Reviewer 4 Report
Submitted paper presents a research on surface-water interaction, more specifically aquifer recharge. The topic is well known in the field of groundwater hydrology and vadose zone studies; the research applies statistical analysis to research and obtained results. However, many other techniques or tools currently applied in this type of studies (hydrochemistry, isotopes, etc provided by other researchers or from this research.) should be jointly applied with statistics to better assess obtained results. The management-environmental issues presented for the area maybe of interest for international readers.
Round 2
Reviewer 2 Report
The manuscript was improved and it can be published in the present form
Reviewer 3 Report
Sorry,I insist that this paper lacks novelty.
Reviewer 4 Report
Can be accepted. I also suggest to include in abstract results from discussion on adaptive management issues